# Autonomous Hazardous Gas Detection Systems: A Systematic Review

**DOI:** 10.3390/s25216618

**Published:** 2025-10-28

**Authors:** Boon-Keat Chew, Azwan Mahmud, Harjit Singh

**Affiliations:** 1Faculty of Artificial Intelligence and Engineering, Multimedia University, Cyberjaya 63100, Selangor, Malaysia; 1221400082@student.mmu.edu.my; 2HyveGeo, London W1W 5PF, UK; harjit.singh@btinternet.com

**Keywords:** sensor calibration techniques, cross-sensitivity, autonomous calibration, gas sensors, calibration drift, machine learning, autonomous systems, gas detection algorithms, multivariate analysis, autonomous calibration

## Abstract

Gas Detection Systems (GDSs) are critical safety technologies deployed in semiconductor wafer fabrication facilities to monitor the presence of hazardous gases. A GDS receives input from gas detectors equipped with consumable gas sensors, such as electrochemical (EC) and metal oxide semiconductor (MOS) types, which are used to detect toxic, flammable, or reactive gases. However, over time, sensors degradations, accuracy drift, and cross-sensitivity to interference gases compromise their intended performance. To maintain sensor accuracy and reliability, routine manual calibration is required—an approach that is resource-intensive, time-consuming, and prone to human error, especially in facilities with extensive networks of gas detectors. This systematic review (PROSPERO on 11th October 2025 Registration number: 1166004) explored minimizing or eliminating the dependency on manual calibration. Findings indicate that using properly calibrated gas sensor data can support advanced data analytics and machine learning algorithms to correct accuracy drift and improve gas selectivity. Techniques such as Principal Component Analysis (PCA), Support Vector Machines (SVMs), multivariate regression, and calibration transfer have been effectively applied to differentiate target gases from interferences and compensate for sensor aging and environmental variability. The paper also explores the emerging potential for integrating calibration-free or self-correcting gas sensor systems into existing GDS infrastructures. Despite significant progress, key research challenges persist. These include understanding the dynamics of sensor response drift due to prolonged gas exposure, synchronizing multi-sensor data collection to minimize time-related drift, and aligning ambient sensor signals with gas analytical references. Future research should prioritize the development of application-specific datasets, adaptive environmental compensation models, and hybrid validation frameworks. These advancements will contribute to the realization of intelligent, autonomous, and data-driven gas detection solutions that are robust, scalable, and well-suited to the operational complexities of modern industrial environments.

## 1. Introduction

Semiconductor wafer fabrication (fab) facilities are renowned for their cutting-edge technologies and ultra-clean, tightly controlled environments. These facilities are also highly process-intensive and involve the use of specialty gasses that are flammable, combustible, and hazardous to both human health and the environment. Gas Detection Systems play a critical role in the monitoring of potential gas leaks to ensure safety and regulatory compliance [1]. The primary functions of GDSs include initiating audio and visual alarms to warn personnel of potential gas hazards, triggering corrective actions, such as process shutdowns, and isolating the hazard from human exposure. The system also provides an alarm history log and graphical trending tools for incident investigation and risk mitigation [2].

The functionality of GDSs depends on gas detectors equipped with consumable sensors that physically interact with the target gas. When exposed to the target gas, these sensors generate millivoltage or milliampere signals, or alter circuit resistance, depending on the specific sensor technology employed. A sample of GDS screen is shown on Figure 1.

Common sensor types include electrochemical, metal oxide semiconductor, catalytic, non-dispersive infrared, and photo-ionization diode sensors [3]. However, these sensors can also respond to interference gases with similar chemical characteristics, which can trigger false alarms, cause unnecessary evacuations, and cause costly production shutdowns.

Current industrial practice for maintaining gas detector performance is based on routine manual calibration performed by trained professionals. The standard manual gas detector calibration process entails a trained professional utilizing a portable calibration gas cylinder, injecting the gas into the detector, and subsequently adjusting the detector’s reading to align with the known concentration of the calibration gas. In a typical wafer fab facility, there may be hundreds or thousands of gas detectors, making the calibration process highly resource-intensive. Reducing the number of installed detectors is generally not viable because of the complexity and diversity of the specialty gases used.

The industry’s drive to reduce costs and improve productivity and safety has spurred research into alternative sensing strategies. One prominent area of investigation is the application of low-cost gas sensors. These sensors have shown significant advantages in other domains, such as ambient air pollution monitoring, where they have allowed reduced monitoring costs and improved spatial coverage [4]. However, they are not a panacea; the calibration of low-cost sensors remains a formidable challenge, with persistent issues of poor selectivity and long-term instability being well-documented barriers to their reliable deployment [2,5,6].

This challenge extends to more sophisticated systems like the electronic nose (e-nose), a device composed of a gas sensor array coupled with machine learning algorithms to mimic the human sense of smell. Although e-nose systems show great promise, their application has been largely confined to laboratory settings, falling short of fulfilling demanding industry requirements. A significant obstacle that has hindered their widespread adoption is the difficulty of “calibration transfer”: ensuring that a calibration performed on one device remains valid when transferred to another, a critical requirement for scalable deployment [7].

It is clear that the need for frequent and resource-intensive calibration is a fundamental and pervasive challenge across the landscape of gas-sensing technology, from established industrial systems to emerging low-cost and intelligent platforms. This systematic review, therefore, explores existing approaches aimed at minimizing gas sensor calibration requirements while maintaining the accuracy and reliability of gas detectors. The objective is to identify and analyze innovative methodologies that can reduce the dependence on manual intervention, paving the way for more autonomous, cost-effective, and robust gas detection solutions.

## 2. The Landscape of Gas Hazards in Semiconductor Wafer Fabrication Facilities

The pristine and highly controlled environment of a wafer fab as shown in Figure 2 belies the significant chemical hazards inherent in its core processes.

Phosphine is used in semiconductor manufacturing, and its handling requires careful attention due to its toxicity. The damaged gas room shown in the figure highlights the risks involved.The use of numerous specialty gases, many of which are pyrophoric, toxic, or corrosive, creates a high-risk environment in which failures in containment or monitoring can have catastrophic consequences.

### 2.1. The Nature of Hazardous Specialty Gases

The range of specialty gases used in wafer fabrication facilities is extensive, with a hazard profile that requires extreme caution, as shown in Figure 3. The list is not exhaustive; environmental regulations often drive the introduction of new, and sometimes equally hazardous, replacement chemicals. For example, chlorine trifluoride (ClF_3_) was introduced to the semiconductor industry in the 1990s as a replacement for perfluorocompounds (PFCs) after the latter were targeted by the Kyoto Protocol [8]. ClF_3_ is an exceptionally dangerous substance, classified as an Extremely Hazardous Substance (EHS) by the National Research Council (NRC) [9]. A notable incident in 1955 involved the spillage of one ton of ClF_3_, which reportedly burned through 30 cm of concrete and 90 cm of gravel while releasing highly corrosive and toxic hydrofluoric acid vapor [10].

Silane is another ubiquitous and extremely hazardous gas, used since the 1950s for its high reactivity in Chemical Vapor Deposition (CVD) processes [12]. Its use has been associated with numerous catastrophic industrial incidents. In 2007, a major explosion and fire in a silane gas room in a photovoltaic module plant resulted in one fatality and total destruction of the gas room, with the fire spreading into the clean room and causing significant damage to the facility [13].

In the same year, a silane explosion in India resulted in a death in which a worker was decapitated and propelled through a brick wall [13]. A further incident in Taiwan in 2005 involved a spontaneous silane explosion during a routine procedure that killed a worker and shut down factory production for three months [14]. Figure 4 shown the Damage to a gas room following a silane explosion.

### 2.2. Process-Specific Risks and Chemical Hazards

Inevitable risks are deeply embedded in the core fabrication processes. Chemical Vapor Deposition (CVD), a process for depositing thin films on silicon wafers as per Figure 5a, involves high temperatures and reactants like silane (SiH_4_), ammonia (NH_3_), and the highly toxic phosphine (PH_3_) [15]. Silane’s thermal decomposition allows for the controlled deposition of silicon-containing layers; for instance, its reaction with oxygen (SiH_4_ + O_2_ → SiO_2_ + 2H_2_) is fundamental for creating insulating layers [15]. Phosphine is used for doping silicon, releasing elemental phosphorus upon decomposition to enhance electrical conductivity. Its oxidation reaction is given by:(1)$$4PH_{3}+5O_{2}⟶2P_{2}O_{5}+6H_{2}$$
This reaction produces phosphorus pentoxide and hydrogen [16]. While crucial for device fabrication, phosphine is both highly toxic and flammable.

Other processes introduce further hazards. Chemical etching often uses solutions based on hydrofluoric acid (HF) to selectively remove materials like silicon dioxide (SiO_2_) or phosphosilicate glass (PSG) as shown in Figure 4a [17]. Storage and delivery of dopant gases such as arsine, phosphine, and boron trifluoride for ion implantation also present significant environmental, health, and safety challenges [18]. Furthermore, process by-products can create latent hazards. Materials such as aluminum chloride (AlCl_3_) or ammonium hexafluorosilicate ((NH_4_)2(SiF_6_)) can condense in exhaust pipes as shown in Figure 5b [19]. These condensates may contain partially reacted hazardous materials that can react violently during subsequent processes, such as chamber cleaning with fluorine gas, potentially igniting fires and causing severe equipment damage [19].

**Figure 5 sensors-25-06618-f005:**
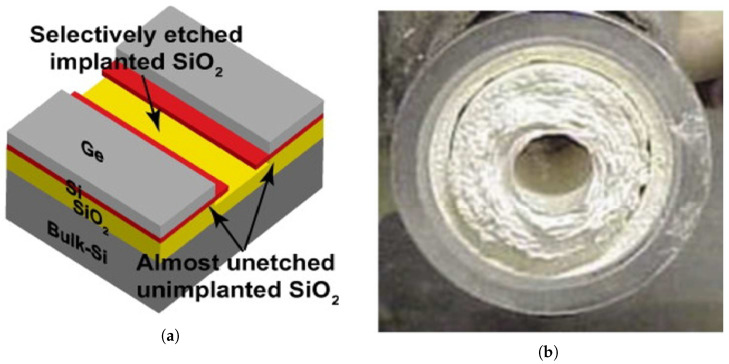
(**a**) (Schematic of selective etching of implanted SiO_2_ [20]. (**b**)By-product condensation in a process exhaust pipe [19].

Finally, the delivery of these specialty gases from storage to production tools is an inherently dangerous operation. Since the 1980s, safety and industrial hygiene authorities have expressed serious concern about the risks posed by even minor leaks of gases such as arsine, phosphine, silane, and hydrogen chloride, imposing greater scrutiny on the industry [21]. These examples underscore the absolute necessity for continuous and reliable gas monitoring systems, the performance of which is the central focus of this review.

### 2.3. Gas Sensor Technologies and Current Calibration Practices

There are a wide selection of sensor technologies applied in different applications. A specific type of sensing film used in e-nose systems involves a two-step fabrication process with carbon black dispersed in a solution containing a dissolved insulating polymer before dispersing on a ceramic substrate [22]. The evaporation of the solvent forms conductive pathways within the polymer matrix, making the films electronically conducting. When a vapor is sorbed, the sensing film swells, breaking the contact between carbon particles and thus increasing the resistance of the sensing layer [22]. Although other complex mechanisms also affect resistance, such as carbon sorption and ionization of the analyte in trapped water, changes in sensor resistance are indicative of changes in the vapor constituents of the environment [23]. The amplitude of the signal variation and the dynamics of these sensors depend on the composition of the sensing layer and the stimulus presented, and the collection of these responses provides a multivariate response that contains a fingerprint of each component of the gas sample [24,25].

Most of the gas sensors used in semiconductor wafer fab are metal oxide semiconductor sensors and electrochemical sensors. The metal oxide semiconductor sensor as per Figure 6 operates on an oxidation and reduction principle. When the sensor is energized by direct current (DC), the sensor surface is anodized and attracts the oxygen ion. The process is also known as “sensor aging”, which is important for the sensors to function properly. Once the sensor is properly aged, the circuitry resistance increases. MOS sensors are based on the adsorption of gas molecules to the surface of the sensor material and the capture of electrons [26]. The electrochemical sensor shown in Figure 7 measures gas concentrations by detecting the electrical current generated from a redox reaction. When the target gas interacts with an electrode, oxidation occurs at the working electrode and reduction occurs at the counter electrode. This process creates an ion flow that produces a current directly proportional to the gas concentration.

When comparing the response time between the EC and MOS sensors, MOS sensors have a faster response time, but are prone to cross sensitivities of solvent vapors [27]. Although semiconductor sensors have been used for NO_2_ detection [28], they suffer from poor gas selectivity and can react with other gases. Although electrochemical sensors are often used in controlled environments, detecting low NO_2_ concentrations near the 3 ppm threshold limit value [29] is problematic due to interference from coexisting gases. These limitations in selectivity and interference make accurate low concentration NO_2_ detection challenging for both sensor types [30]. With the help of filters, metal oxide semiconductor gas sensors could become quite selective to some targeted gas such as methane [31]. In order for the gas detector to function according to the manufacturer’s design standard, periodic maintenance is required, such as sensor replacement and calibration. Routine calibration and bump test can check the gas sensor depletion rate, which is subjective to the application and environment where the gas detectors are installed.

During the calibration process, a standard calibration gas is injected into the gas sensor to simulate a gas detection. The reading on the gas detector will be adjusted according to the unit reading of the calibration gas. Gas detectors from different manufacturers have different performance specification, maintenance procedure, and characteristic. As part of the calibration procedure, the response time of the gas detectors will be tested after each calibration, which is also known as the bump test. A bump test is conducted to test the T90 response time, which is a test to verify the time it takes for the gas detector to show 90 percent of the full range of gases. A proper bump test is a good indication of the gas sensor condition. A general rule of thumb for the T90 bump test is 30 s or less for combustible gas and 60 s or less for toxic gas. When the bump test result indicated any value above the aforementioned value, it is a good indication for sensor replacement.

Calibration is critical for ensuring the accuracy of gas detectors, directly impacting the overall performance of the GDS in detecting hazardous gases, reducing false alarms, and preventing industrial accidents. The challenge lies in executing timely and consistent calibration across wafer fabrication facilities where a single fab may be installed with close to a thousand gas detectors. These detectors may come from different manufacturers, employing various sensor technologies that require different calibration procedures.

This reliance on manual calibration is a critical vulnerability. The process is labor-intensive and time-consuming and requires highly trained personnel to adhere to strict schedules. In a large facility with nearly a thousand detectors from different manufacturers, ensuring timely and consistent calibration is a significant logistical and financial challenge. More importantly, it introduces the risk of human error. Studies have long identified human error in maintenance as a pressing industrial problem and a leading cause of accidents [32]. Analysis of major industrial disasters, such as those in Chernobyl and Bhopal, has shown that the “human factor” is a crucial contributor in up to 90 percent of incidents [33]. In the context of gas detector calibration, worker fatigue can lead to mistakes, compromised consistency, and overlooked anomalies, directly undermining the integrity of the safety system. Figure 8 outlines the inherent issues associated with gas sensors, which pose additional challenges for manufacturing plant owners, regardless of whether they opt for sensor calibration or choose not to calibrate.

Early research has explored methods to address these calibration challenges. For example, the operation of sensors with a temperature cycle has been shown to limit cross-sensitivities and aging effects under laboratory conditions [34,35]. The use of artificial neural networks for field calibration has also been investigated, although with mixed results, proving satisfactory for short periods but generally weak for longer data series [36,37]. These initial studies highlight the complexity of the problem and underscore the need for more robust and reliable solutions.

## 3. Methodology

This systematic review (PROSPERO on 11 October 2025 Registration number: 1166004) was designed and reported according to the Statement of Preferred Reporting Items for Systematic Reviews and Meta-Analyses (PRISMA) 2020. The PRISMA framework provides a robust and transparent methodology for literature reviews, ensuring that the processes of study identification, selection, evaluation, and synthesis are explicit and reproducible. A protocol for this review was established a priori to define the objectives, search strategy, and analysis methods before the review began, thus minimizing bias.

Eligibility criteria: Studies were selected for inclusion based on a predefined set of criteria structured around the Population, Intervention, Comparison, and Outcome (PICO) framework.

Population: Gas detection systems, including individual sensors, sensor arrays, and e-noses, are used specifically in air quality monitoring and related to human health and safety.

Intervention: Methodologies, algorithms, machine learning, or technologies designed to reduce, minimize, or eliminate the need for traditional periodic manual calibration. This also includes, but is not limited to, drift compensation algorithms, self-calibration techniques, calibration transfer methods, and algorithms to differentiate target gases from interference gases.

Comparison: Comparisons are made between experiments and studies in the development of methodologies, algorithms, machine learning from the collection of gas-sensing data in electronic noses, air quality monitoring, and industrial gas-detection applications.

Outcomes: The primary outcomes of interest are the performance and reproducibility of the calibration model and the reliability of the methodology, algorithm, and machine learning in differentiating the target gas from the interference gases. This includes metrics such as precision, selectivity, stability, response time, and effectiveness of the intervention to mitigate sensor drift or performance degradation over a specified period.

### 3.1. Information Sources and Search Strategy

A comprehensive search strategy was developed to identify all relevant studies, including both published and unpublished “grey literature” to mitigate publication bias. The search was carried out on multiple electronic databases relevant to engineering, chemistry, and computer science, mainly IEEE Xplore, Scopus, and MDPI. The search strategy combined keywords and database-specific subject headings (e.g., MeSH terms) related to the core concepts of the review. Key search concepts included “gas sensor”, “chemical sensor,” “e-nose,” “calibration”, “drift compensation”, “self-calibration,” and “calibration transfer” and “AI” and “machine learning”. These terms were combined using Boolean operators (“AND,” “OR”) to achieve high sensitivity. In addition to database searching, supplementary methods were employed, including snowballing (reviewing the reference lists of included articles) and hand-searching key journals in the sensor technology field.

### 3.2. Study Selection and Data Extraction

The study selection process was conducted in two phases. In the first phase, the titles and abstracts of all retrieved records were screened for relevance against the eligibility criteria. In the second phase, the full texts of potentially relevant articles were recovered and evaluated for final inclusion. The entire selection process is documented and presented in a PRISMA 2020 flow diagram as per Figure 9, which visually depicts the flow of information and details the reasons for the study exclusion. The key data points extracted included the study author and year, study design, sensor type, application environment, details of the calibration-minimizing intervention, performance metrics, and main findings.

### 3.3. Synthesis of Results

Given the anticipated heterogeneity in sensor technologies, experimental protocols, and performance metrics across included studies, it is unlikely that a statistical meta-analysis is appropriate. Therefore, the findings will be synthesized using a narrative synthesis approach. This method involves the use of text, tables, and flow diagrams to summarize and explain the findings of the included studies. The synthesis will be structured thematically, grouping studies based on the type of intervention (e.g., algorithmic drift compensation, novel material development, calibration transfer techniques). This approach will allow the exploration of relationships and patterns within and between studies, providing a comprehensive overview of the current state of research and identifying key trends, challenges, and promising future directions.

## 4. Comprehensive Systematic Review

The advancement in gas sensor technologies over the past decade has addressed some key challenges such as accuracy drift, sensor degradation, and environmental effects. This section reviews the selected literature related to gas sensor data collection with an emphasis on specific applications, sensor data analytics for accuracy enhancement, and early fire hazard detection using algorithms in differentiating the target gas.

### 4.1. Sensor Accuracy Drift and Dataset Limitations

The reliability of gas sensor systems is fundamentally challenged by sensor drift, which degrades accuracy and limits long-term deployment. Sensor drift manifests itself as gradual, non-deterministic temporal variations in a sensor response, even when exposed to the same analyte under identical conditions [39]. The degradation is caused by a combination of aging and poisoning of impurities, and changes in temperature and humidity [23,40,41]. The resulting decay in sensor selectivity and sensitivity can render pattern recognition models obsolete in a matter of months, necessitating complete and costly system recalibration [42]. These challenges are particularly acute in open sampling systems (OSSs), where sensors are directly exposed to an environment in which the dispersion of gases through diffusion, turbulence, and advection makes reliable identification difficult [23]. OSS deployments are also prone to sample contamination, measurement time delays, and system failures resulting from improper design that can compromise data integrity [1]. A prominent case study that highlights these issues involves a comprehensive dataset from a chemical gas sensor array deployed in a turbulent wind tunnel to simulate the complexities of OSS. In total, 16 chemical sensors were used to gather 13,910 measurements of six different gaseous substances: ammonia, acetaldehyde, acetone, ethylene, ethanol, and toluene over 16 months, creating a valuable resource to study sensor variability [43]. Subsequent analysis of this widely used dataset revealed critical limitations for gas classification benchmarks [23,44]. Instead of being measured in random order, gases were recorded in separate, time-separated batches, introducing a significant long-term drift that correlated with the measurement time [45]. This finding renders the dataset unsuitable for benchmarking gas recognition algorithms without drift correction and suggests that the performance of the algorithms in at least 18 publications that used this dataset may have been significantly overestimated [23]. This underscores the critical need for both drift-resistant algorithms and new, carefully designed datasets that avoid such methodological pitfalls. The challenge of sensor drift is multifaceted and involves distinct components that require different approaches. Research distinguishes between “instrumental drift” arising from physical changes in the sensor and “environmental variation” caused by external factors. A related concept is “concept drift,” which refers to changes in the underlying statistical properties of data over time, making models trained in one context less effective in another [46]. Failing to properly distinguish and account for these different drift sources can lead to improper calibration and flawed performance evaluations, as standard cross-validation techniques often overestimate the accuracy of a model in the real world by not adequately accounting for temporal drift [23,47].

To better address the complexities of real-world conditions, other comprehensive datasets have been developed. One such effort involved the deployment of a large array of 72 MOX sensors in a turbulent wind tunnel to simulate an OSS environment. This experiment captured 18,000 measurements of 10 volatile compounds under varying operating temperatures and wind speeds over 16 months, providing a rich resource for studying sensor variability and gas plume dynamics in challenging settings [48].

Building on these datasets, research has focused on developing advanced analytical techniques. In one study, Inhibitory Support Vector Machines (ISVMs) were applied to discriminate turbulent gas mixtures, achieving a precision of more than 97 percent in identifying the presence of ethylene when trained in a full range of concentrations [49]. This work highlights the necessity of using low-concentration training data for robust classification in open systems. Another novel methodology used Mutual Information (MI), an information–theoretic metric, to systematically optimize the operating temperatures of a full MOX sensor array for a given classification task, moving beyond previous approaches that focused only on single-sensor optimization [50]. The methodologies is summarized in Figure 10.

Additional studies revealed critical limitations for gas classification benchmarks due to sensor drift. The studies showed that instead of being measured in randomized order, the gases were recorded in time-separated batches, introducing long-term drift correlated with measurement time. Principal Component Analysis (PCA) and Support Vector Machine (SVM) classification showed that baseline signals alone allowed near-perfect gas identification (94.3 percent precision) [44]. A restricted subset (methanol, ethylene, butanol) was proposed from temporally proximate measurements, achieving approximately 60 percent accuracy. This work underscores the unsuitability of the dataset for gas recognition benchmarks without drift correction and urges the development of drift-robust algorithms and new datasets. Furthermore, most studies on neural network calibration have used only low-cost MOx-type sensors, which are known to suffer from a lack of stability and a long response time [51].

The challenge of calibration also extends to large-scale environmental monitoring. A study in the Puget Sound region deployed a network of 54 low-cost monitors (LCMs) equipped with electrochemical sensors to measure CO, NO, NO_2_, and O_3_ over nearly two years [52]. Using multiple linear regression (MLR) models that incorporated sensor signals, temperature, humidity, and co-pollutant data, the study demonstrated practical calibration strategies to improve long-term pollutant exposure predictions, underscoring the potential of calibrated low-cost sensor networks to enhance regulatory monitoring [52].

### 4.2. Calibration Methodologies for Low-Cost Sensors

The viability of low-cost sensors for large-scale air quality monitoring systems (AQMSs) is an active area of research, with the potential to create hybrid networks that supplement conventional analyzers and address data sparsity [53,54,55,56]. However, their widespread adoption is hindered by significant challenges. Long-term stability is an important concern due to measurement drift, which may be addressed by adaptive or semi-supervised calibration schemes [23,57]. Furthermore, poor sensor-to-sensor reproducibility complicates the use of a single calibration function, requiring research on calibration transfer strategies to make large-scale deployment feasible [58]. The calibration methodologies and objectives are listed under Table 1.

These challenges are particularly evident in electronic nose (e-nose) systems, which are used in numerous domains [59,60,61,62,63]. An e-nose relies on a prediction model learned from training samples, but its universality is limited by manufacturing variations, aging, and poisoning of the gas sensors [64,65,66]. These factors cause the data distribution of test samples (the target domain) to differ from the original training samples spectroscopic data [67,68,69], affecting model performance. To address this, various algorithmic approaches have been developed, many falling under the umbrella of “calibration transfer” [7,67,68,70,71,72,73]. Sensor drift is unpredictable in practical applications, making drift compensation more difficult [74].

Traditional methods often focus on signal preprocessing, such as baseline manipulation or drift component correction using techniques such as orthogonal signal correction (OSC) and component correction using principal component analysis (CCPCA) [75,76]. More advanced solutions leverage machine learning. Transfer learning frameworks aim to adapt knowledge from the source to the target domain, with methods like Direct Standardization (DS) using a small set of transfer samples to map a new “slave” sensor’s signals to the original “master” sensor’s space [77]. A study applying DS to MOX sensor arrays demonstrated a significant reduction in prediction errors and the ability to mitigate drift over time [77]. Other approaches, such as multi-task learning (MTL), learn models for both domains simultaneously, sharing information to improve accuracy on drifted data [78,79]. Principal Component Analysis (PCA) was considered a regular method for further related studies, and Independent Component Analysis (ICA) is another method with a stricter decomposition rule to capture drift signals [80]. Research on ensemble classifiers has focused on modifying mathematical models using diverse labeled datasets [81,82].

Sensor arrays are distinctive in artificial chemical sensing, drawing inspiration from the combinatorial selectivity principles observed in natural olfaction [83]. Sensor arrays consist of widely cross-reactive chemical sensors, each generating unique chemical signature co-relating to a diverse range of analyte compounds. The data for multi-chemical sensor arrays is generally segmented into feature extraction and classification model. Feature extraction identifies a comprehensive set of attributes associated with the sensor response signals that are pertinent to the specific application [84,85,86,87,88] to maximize performance [89,90,91]. Classification model procedures investigate the use of linear or nonlinear pattern recognition techniques (generally classifiers) to develop robust and reliable inferences of the properties of the measured sample [92,93,94,95]. Although considerable progress has been made toward delineating how calibration curves are transferred among different arrays of the same kind [70,96,97], the possibility of achieving these attributes by fine-tuning sensors is still rather limited, especially for applications in real confounding atmospheres. To mitigate the inherent variability among sensor units, a multi-unit calibration methodology was introduced to develop generalized models that are not dependent on any specific sensor replica. By utilizing data from multiple sensor arrays to train Partial Least Squares Discriminant Analysis (PLS-DA) models, researchers created robust algorithms that obviate the necessity for individual unit calibration [98]. In a novel approach to eliminate the need for calibration gas entirely, a study designed a self-calibrating electrochemical sensor that uses its own electrical impedance to estimate the active electrochemical area, which correlates with sensitivity and response time, enabling online functionality monitoring [99].

### 4.3. Application-Specific Deployments and Environmental Variable Studies

The performance of low-cost sensors is heavily influenced by their deployment environment. Several studies have focused on calibrating sensors for specific applications while accounting for environmental variables as listed in Table 2. In one study, the integration of electrochemical sensors with unmanned aerial vehicles (UAVs) for NO_2_ monitoring required a novel calibration approach that incorporated the rotor speed of the UAV to mitigate its interference with sensor readings, significantly improving the accuracy of the measurement [100]. A review of sensors for monitoring benzene and other volatile organic compounds in ambient air found that while some technologies, such as Photoionization Diode, could achieve the necessary sub-ppb detection limits, they often lacked selectivity without specialized filters [101]. In mobile robotics, a MOX sensor array was integrated into an Assistant Personal Robot (APR) for gas leak detection, using a PLS-DA classifier to distinguish between different vapors in complex indoor settings [102]. Other applications include smart sensor arrays to detect gases related to spoilage in food packaging [103] and to characterize the influence of gas flow rate on sensor performance for industrial leak detection [104].

The impact of temperature and humidity is a recurring theme. A study of Alphasense electrochemical sensors confirmed that, while the sensors showed excellent linearity under stable conditions, variations in temperature and humidity significantly altered their sensitivity and baseline outputs [105]. Another study demonstrated a method to correct for this by extracting daily temperature-dependent baselines from high-resolution data and fitting them to exponential or linear functions, which, when subtracted from the raw signal, greatly improved correlation with reference measurements [106]. Similarly, a study in a tropical environment in Sri Lanka found that frequent field calibrations (every 3 to 6 months) were necessary to counter the strong effects of high temperature and humidity on the performance of the CO sensor [107].

The specific application also dictates the required sensor performance. Several studies have explored the performance of low-cost gas sensors in real-world deployments. A network of low-cost electrochemical and metal oxide sensors was installed to monitor NO_2_ along the A22 Brenner road in Italy within the BrennerLEC project, with the aim of assessing traffic-related pollution and impact on speed limits [108]. Ten AirQino (AQ) sensors were deployed, with dual calibrations (summer and winter) against reference stations, testing multivariate regression (MR), temperature-dependent MR (T-MR), spline interpolation (SPL), and random forest (RF) models. A year-long validation of AQ1 and AQ3 versus reference stations (ML103, BL164) showed good agreement with better winter performance due to stable conditions and proximity of calibration to highway conditions. This study demonstrates the reliability of low-cost sensors for long-term roadside monitoring, improving spatial coverage, and supporting the validation of dispersion models, although expert management is needed to maintain accuracy of calibration.

The calibration of 10 units of Alphasense CO-B4 electrochemical sensors in conjunction with a reference NDIR CO analyzer for CO monitoring in a tropical environment (Kandy, Sri Lanka) addressed temperature (20–32 °C) and relative humidity RH 45–90 percent of effects on sensor sensitivity [107]. Linear regression (SLR) and multiple linear regression (MLR) models, incorporating temperature and RH, were developed and compared against factory calibration. The models show exponential or polynomial temperature dependence with notable variations above 28 °C and 75 percent RH. The study underscores the need for frequent field calibrations (3 to 6 months) in humid and warm climates to counter environmental effects and sensor degradation, improving the low-cost sensor reliability for monitoring urban air quality in tropical settings.

Photoionization detectors (PIDs), electrochemical (amperometric) sensors, metal oxide (MOx) sensors, optical sensors, portable gas chromatographs, and sensor arrays were studied focusing on sensitivity, selectivity to monitor benzene, and volatile organic compounds (VOCs) in ambient air, with the aim of complying with the limit value of 5 µg/m^3^ from the European Air Quality Directive (AQD)(1.5 ppb) [101]. PID sensors from various manufacturers achieved sub-ppb benzene detection limits (LD, 0.5–1 ppb) but lack selectivity without filters. Amperometric sensors are insufficient for ambient air with cross-sensitivity to VOCs and humidity dependence. MOx sensors typically exceed 100 ppb LoD, though marred by noise issues per US-EPA tests. The literature highlights prototypes such as SiC-FET and temperature-cycled MOx, achieving sub-ppb sensitivity with enhanced selectivity via neural networks.

A comprehensive analysis of smart sensor arrays for gas detection in food packaging applications emphasized the importance of detecting spoilage-related gases, such as ammonia, hydrogen sulfide, and VOCs, in modified atmosphere packaging (MAP) to ensure food quality and safety [103]. The authors explore various gas detection technologies including metal oxide semiconductors, conducting polymers, and optical sensors, and highlight the role of data processing, sensor fusion, and wireless integration for real-time monitoring. Challenges like sensor drift and selectivity are addressed, along with solutions involving machine learning and multisensor systems. The review also discusses trends toward miniaturization, low power consumption, and multi-gas detection capabilities, underlining the significance of smart sensor arrays in extending shelf life and reducing food waste.

The flow rate influences the performance of gas sensors based on microheaters, particularly in wireless monitoring systems [104]. The study uses both simulation and experimental approaches to evaluate how placing obstacles of different sizes and shapes near the sensor reduces the effect of gas flow rate on sensor accuracy. The results show that larger and optimally shaped obstacles, such as quadrangular prisms, significantly minimize voltage response variation caused by flow rate changes. In addition, thermostatical control is used to isolate the direct and indirect effects of gas flow on sensor temperature and chemical reactions. This enables the correction of gas concentration measurements on the basis of flow rate. The proposed techniques improve sensor stability in dynamic environments and suggest the integration of low-power flow, temperature, and gas sensors for more accurate and energy-efficient wireless gas leak detection systems, particularly in industrial applications.

### 4.4. Early Fire Detection Using Gas Sensor Data

The early detection of fire hazards through gas sensing technology and the application of an algorithm to classify the targeted gas have attracted significant research interest. An integrated multisensor prototype comprising an 8-MOX array (AMS), PID (Alphasense), NDIR $CO_{2}$ (Alphasense), electrochemical CO sensor (Alphasense), and humidity sensor (Sensirion) was placed into a 272-L chamber along with commercial smoke detectors for early and reliable fire detection, leveraging the premise that volatile gases precede smoke in many fire scenarios [109]. Experiments involved standardized fires (e.g., pine wood, electrical cables) and nuisances (e.g., ethanol, air freshener), with Partial Least Squares Discriminant Analysis (PLS-DA) used to classify fire versus non-fire scenarios. The results demonstrated the system’s ability to reject nuisances while detecting fires, often triggering alarms faster than smoke detectors (e.g., before photoelectric detectors in some cases).

A similar experiment was conducted by deploying gas sensor arrays involving a prototype with 12 mixed-technology gas sensors in an EN-54 standard fire room (240 m^3^) and a small-scale chamber (0.27 m^3^) targeting smoldering fires that emit CO and VOCs before significant smoke [110]. Data from 52 fire and 27 nuisance experiments were analyzed using machine learning, achieving good classification rates that highlight the potential of gas sensors to detect low-smoke fires.

A review of chemical sensor systems and algorithms for indoor fire detection highlighted their potential for early detection and toxicant monitoring compared to smoke-based detectors [111]. The study notes that gas sensors, including electrochemical CO cells and metal oxide (MOx) sensors, can detect combustible products before smoke, leveraging the fact that toxic emissions, not burns, cause most fire-related casualties. Standards such as EN-54 and UL217 define test fires (e.g., smoldering wood, flaming polyurethane), but gas-based systems lack specific benchmarks.

## 5. The Research Gap and Future Research Direction

### 5.1. The Research Gap

This systematic review reveals that, while significant progress has been made in using low-cost sensors and machine learning to address calibration challenges, several critical research gaps persist. These gaps limit the reliability and large-scale deployment of autonomous gas-sensing systems and point toward clear directions for future research.

A primary issue lies in the generation of datasets, which is the very foundation of algorithmic development. As highlighted, a widely used MOX gas sensor dataset has been shown to have critical limitations for classification algorithm benchmarking due to sensor drift correlated with time, introduced by its experimental design [44]. This finding urges the development of new and methodologically sound datasets that are application-specific and avoid such pitfalls. For example, experiments in turbulent wind tunnels could be improved by using simultaneous measurements from multiple sensor arrays at different locations to better decouple sensor drift from environmental variability.

Furthermore, there is a gap in understanding the fundamental mechanisms of drift itself. Current research often attributes drift to the time elapsed between measurements, but there is little investigation into how accuracy drift is affected by the cumulative amount and duration of gas exposure. This is particularly relevant for MOX sensors, whose operation depends on the accumulation and depletion of oxygen ions on the sensor surface. A deeper understanding of this dose-response relationship with drift is needed.

Another significant gap exists in the validation methodologies used in many studies. Research often compares data from low-cost sensors directly with data from reference gas analyzers without adequately accounting for the distinctive operating principles and sampling methods. Gas analyzers often condition samples (e.g., by cooling to remove moisture), whereas ambient sensors do not. This methodological mismatch means that the measurements from the two systems are not directly comparable, potentially leading to flawed conclusions about sensor performance.

Finally, while considerable research has focused on drift compensation for individual sensors or arrays [83,112,113,114], the problem of calibration transfer between different sensor units has received much less attention. The inability to reliably transfer a calibration model from a master unit to other replica units is a major barrier to the cost-effective large-scale deployment of sensor networks, as it necessitates the expensive and time-consuming individual calibration of every device.

### 5.2. Future Research Direction

#### 5.2.1. Development of Application-Specific Datasets

Future research should prioritize the development of application-specific datasets tailored to the unique demands of semiconductor wafer-fab environments. These datasets must capture the complex behavior of gas sensors in real-world conditions, encompassing diverse background gas compositions, such as nitrogen or trace volatile organic compounds, and environmental variations, such as temperature and humidity fluctuations. There are already many research studies on gas sensors of various technologies confirming the effect of relative humidity and temperature on gas measurement and long-term performance sustainability [107]. A critical area requiring further investigation lies in the development of robust methodologies to address sensor performance degradation and failure. Reflecting the operational conditions of cleanrooms and fabrication facilities, these datasets will enable the creation of robust machine learning models capable of accurately identifying target gases while distinguishing them from interferences, thus enhancing the reliability of gas detection systems. To ensure data consistency and address the critical issue of temporal drift, which undermines the performance of supervised learning algorithms, research should incorporate synchronized and spatially distributed sensor arrays. This approach will facilitate simultaneous data collection across multiple points in dynamic settings, such as gas plumes or airflow systems, minimizing the time-related variability that has limited previous studies conducted in controlled environments such as wind tunnels. By overcoming these temporal uncertainties, the resulting datasets will support the development of drift-resistant algorithms, improving the precision and generalizability of gas classification models. Although various algorithms have been implemented to cope with sensor drift [114], only a modest group of authors have been dedicated to alleviate the presence of faulty sensors [66,113,115,116]. One of the initial works to improve the simulated sensor malfunctions in a five-sensor array proposed a method to identify the damaged sensor and replace its response with an estimate given by the other sensors [117].

#### 5.2.2. Quantitative Modeling of Real World Gas Exposure vs. Sensor Response Drift

Future research should advance the reliability of gas detection systems in semiconductor fabrication by developing comprehensive models that address sensor drift caused by cumulative gas exposure and environmental fluctuations or variables. Current studies often focus on time-based drift, overlooking the critical role of exposure dose (concentration × duration) in sensor degradation, particularly for metal oxide semiconductor (MOS) sensors, where oxidation–reduction dynamics and oxygen ion accumulation drive impedance changes. Research should prioritize modeling the relationship between gas exposure and sensor response drift, capturing how varying exposure patterns, such as intermittent leaks of specialty gases like silane or phosphine, affect long-term performance. This will enable predictive strategies to maintain sensor accuracy in high-risk industrial settings. Simultaneously, the impact of environmental factors such as temperature and humidity variations must be addressed to ensure robust sensor operation. Research can develop holistic frameworks that combine exposure-driven drift models with real-time environmental adjustments, leveraging advanced analytical approaches such as machine learning. These frameworks will improve the precision of gas detection systems, reducing false alarms and calibration needs.

#### 5.2.3. Harmonizing Low-Cost Sensor Outputs with Industrial Grade Benchmark Standards

Future research should enhance the autonomy and reliability of low-cost gas sensors for a specific application by developing frameworks that validate sensor performance against industrial-grade gas sensors calibrated with ISO-standard calibration gases and integrate self-calibration mechanisms. Low-cost sensors, increasingly deployed for their affordability, often lack the precision of their industrial counterparts, creating a critical gap in ensuring accurate detection of gases. Research should focus on hybrid validation frameworks that benchmark low-cost sensor arrays against calibrated industrial sensors under real-world conditions, taking into account factors such as background gas compositions and environmental variations. This approach will ensure that low-cost sensors achieve comparable accuracy, supporting their adoption in safety-critical applications. Research should also explore autonomous calibration strategies to reduce the reliance on labor-intensive manual processes, which are prone to human error and impractical in large-scale facilities. Drawing inspiration from predictive maintenance, self-calibration systems on board and digital twins of sensor arrays can be developed to monitor performance, predict degradation, and dynamically adjust outputs. The digital twins would simulate sensor aging and environmental impacts, anticipating calibration needs to maintain long-term reliability. By integrating validation with self-calibration, these frameworks will enable intelligent gas detection systems that operate autonomously while meeting stringent industrial standards. This direction will drive the development of cost-effective and scalable solutions that enhance safety, minimize false alarms, and improve operational efficiency in semiconductor fabrication, with potential applications in industries such as petrochemicals. By aligning low-cost sensors with industrial benchmarks and enabling self-sustaining operation, future research will support smart factory environments, address the limitations of manual calibration, and foster robust and compliant gas monitoring systems.

The Future Research Gaps and Research Directions are summarized in Figure 11.

## 6. Conclusions

This systematic review identified the gap between experimental research and industrial applicability. Many datasets used for model training failed to replicate real-world conditions. To address these issues, several future research directions are proposed. These include developing application-specific datasets that reflect actual environmental and operational conditions. Deploying sensor arrays for synchronous data collection to reduce time related drift artifacts; modeling the relationship between gas exposure (concentration and duration) and sensor response drift; and implementing adaptive environmental compensation algorithms. Research should focus on bridging the gap between low-cost gas sensors and industrial-grade gas sensors by developing hybrid validation frameworks that condition and normalize sensor data for compatibility. Ultimately, the future of gas detection lies in intelligent data-driven systems that combine sensor technologies with machine learning, robotics, and automation. These systems will not only reduce the reliance on manual intervention, but also enhance safety, operational efficiency, and decision-making capabilities across high-risk industrial domains.

## Figures and Tables

**Figure 1 sensors-25-06618-f001:**
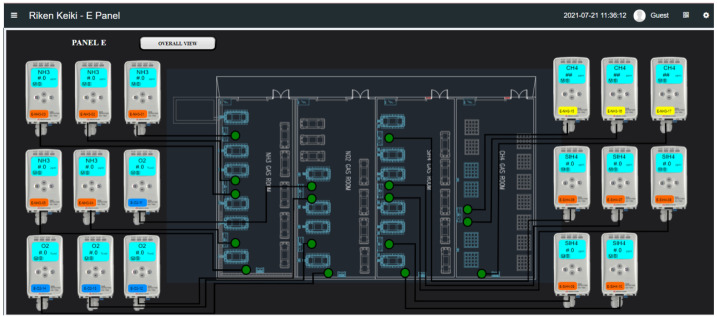
GDS screen indicating alarm status by courtesy of Riken Keiki M Sdn. Bhd.

**Figure 2 sensors-25-06618-f002:**
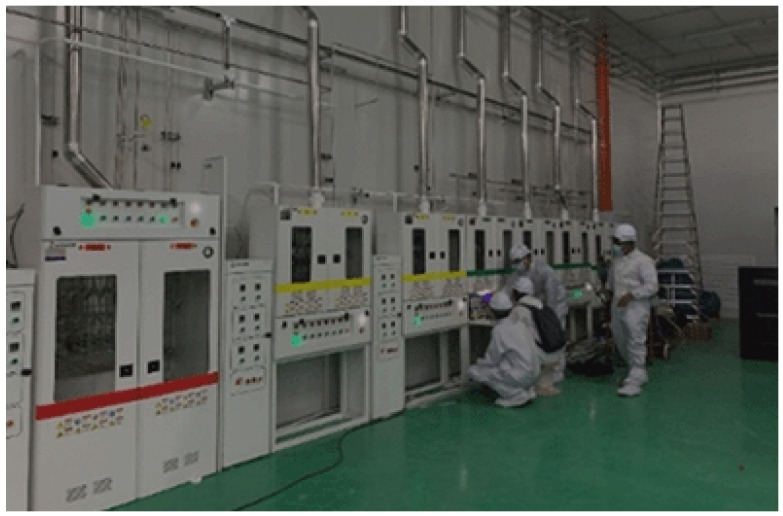
Semiconductor screen Plant Facilities by courtesy of Riken Keiki M Sdn. Bhd.

**Figure 3 sensors-25-06618-f003:**
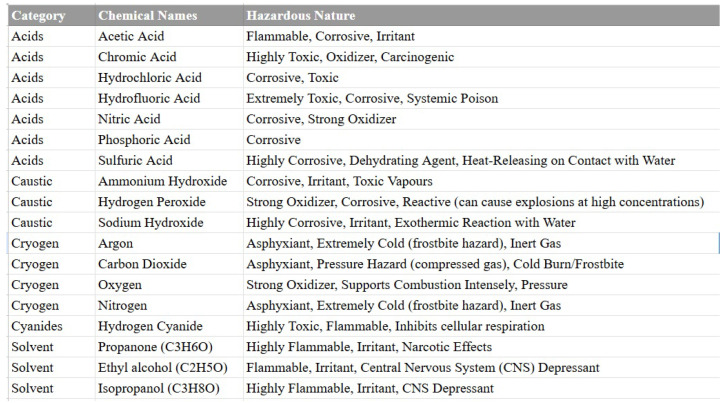
Examples of specialty gas hazards in semiconductor facilities [11].

**Figure 4 sensors-25-06618-f004:**
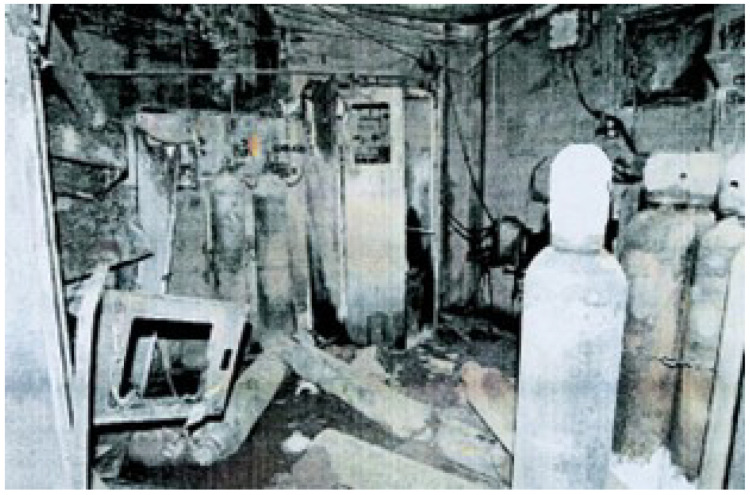
Damages Gas Room by the courtesy of Riken Keiki M Sdn. Bhd.

**Figure 6 sensors-25-06618-f006:**
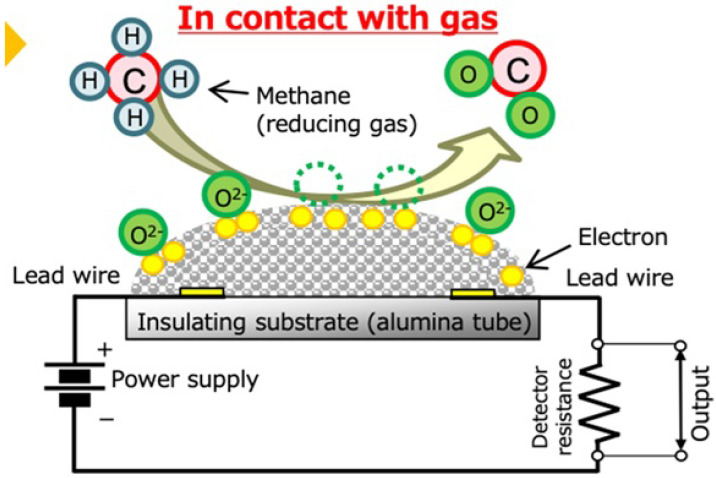
Internal structure of metal oxide sensor by courtesy of Riken Keiki Sdn. Bhd.

**Figure 7 sensors-25-06618-f007:**
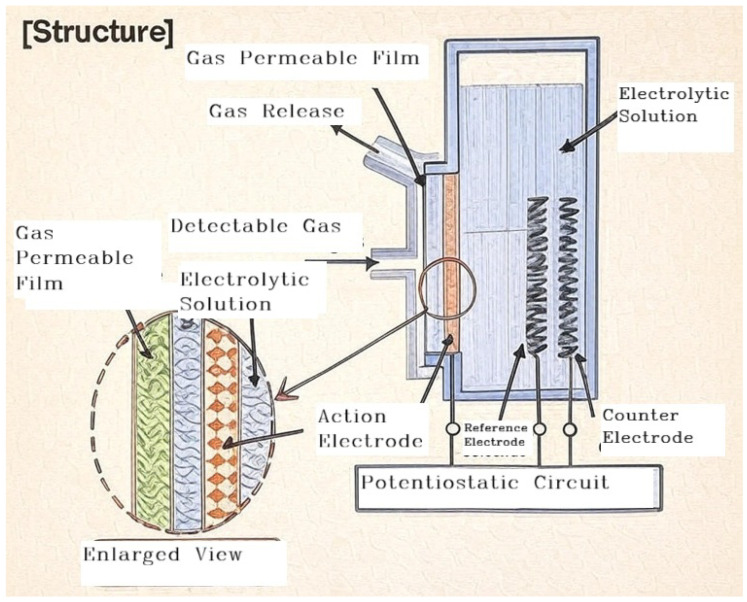
Internal structure of electrochemical sensor by courtesy of Riken Keiki Sdn. Bhd.

**Figure 8 sensors-25-06618-f008:**
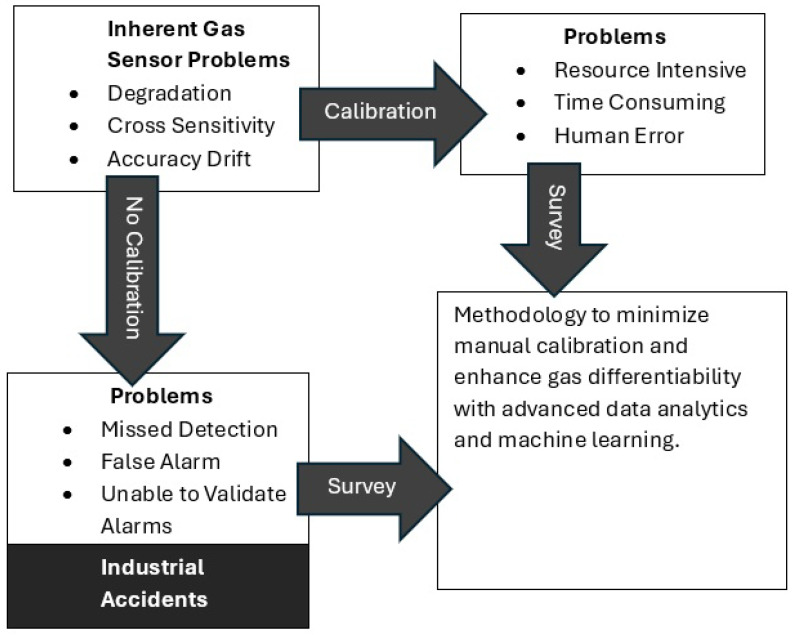
Inherent Gas Sensor Problems and Methodology Using Advanced Analytics.

**Figure 9 sensors-25-06618-f009:**
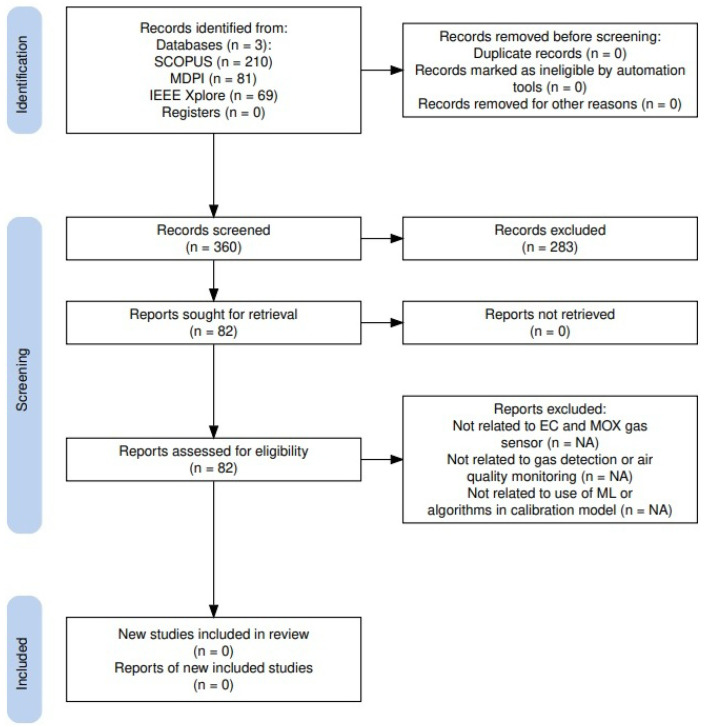
PRISMA 2020 flow diagram [38].

**Figure 10 sensors-25-06618-f010:**
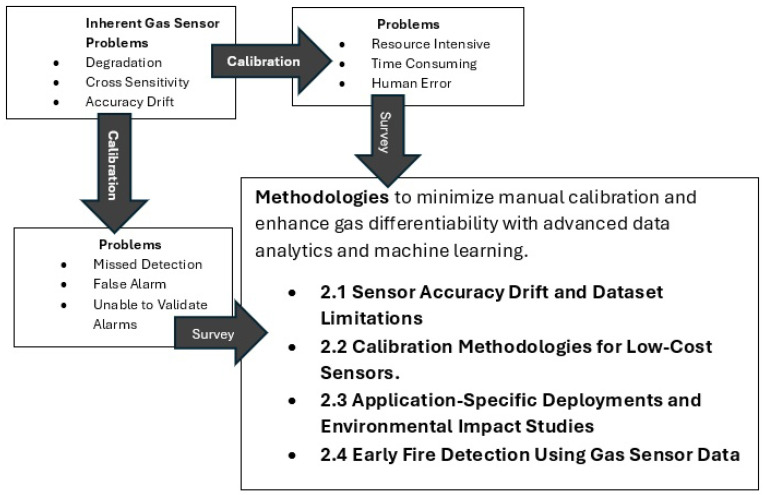
Methodologies to minimize manual calibration and enhance gas differentiability with advanced data analytics and machine learning.

**Figure 11 sensors-25-06618-f011:**
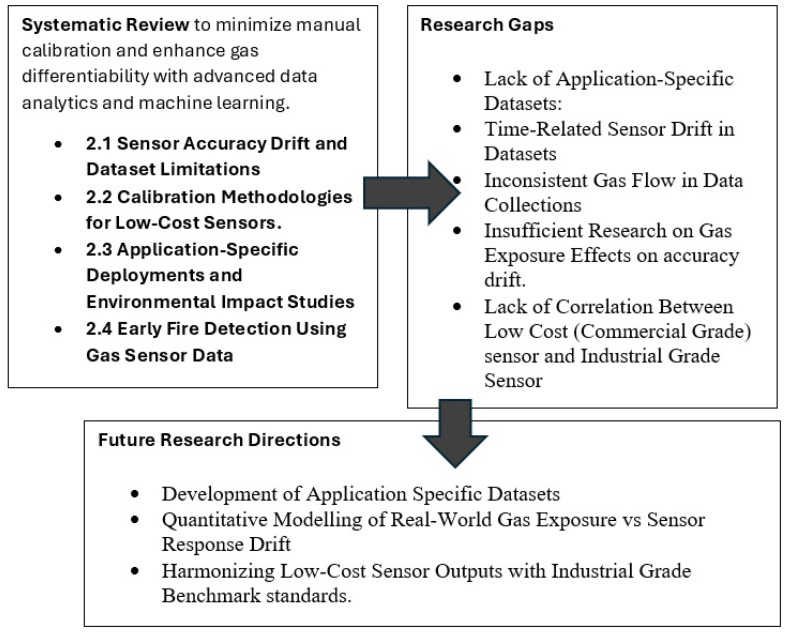
Research gaps and future research directions.

**Table 1 sensors-25-06618-t001:** Calibration methodologies and objectives.

Calibration Methodology	Objectives
Partial Least Squares Discriminant Analysis (PLS-DA) with Genetic Algorithm	To reduce overfitting on general calibration models [29]
Support Vector Machine (SVM) Classification and Support Vector Regression (SVR)	To mitigate accuracy drift due to individual sensors [30]
Correlation of electrical impedance with sensitivity	To monitor sensor aging vs. response time [31]
Correlation of UAV rotor speed with calibration accuracy	To advance UAV gas sensing via a practical calibration method [32]
Correlation of temperature and humidity on temperature-induced baseline study	To study the linearity between gas reading and ambient conditions [33]
	To study the sensor’s long-term stability [34]

**Table 2 sensors-25-06618-t002:** Application-specific deployments and environmental variables impact studies.

Applications	Environmental Variables
NO_2_ monitoring along highway	Pollutant, temperature and humidity [35]
CO monitoring in tropical environment	Temperature and humidity [36]
Ambient air	interference gases, humidity [37]
Gas leak detection with Assistant Personal Robot (APR)	Interference gases [38]
Detect spoilage-related gases in food packaging	Interference gases [39]
Gas detection in air flow	Gas/air flowrate [40]

## Data Availability

Not applicable.

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
