# Peer review of "Autonomous Hazardous Gas Detection Systems: A Systematic Review"

_sensors, 2025, doi:10.3390/s25216618_

Round 1
Reviewer 1 Report
Comments and Suggestions for Authors
This systematic review delves into autonomous hazardous gas detection systems used in semiconductor manufacturing environments. The article points out that traditional gas sensors suffer from accuracy drift, cross sensitivity, and performance degradation, relying on frequent and expensive manual calibration. The author systematically reviewed the research progress on minimizing or eliminating manual calibration through machine learning algorithms, multi-sensor arrays, and drift compensation techniques. And revealed the key gap between research and application. Finally, the article points out the direction for future research. However, there are also some issues in the article that need to be addressed, so my opinion is major revision.
- There are some grammatical and expression issues that require further refinement.
- The conclusion of the abstract mentions 'well suited with', it is suggested to change it to 'well suited to'.
- The introduction mentions' falling short of fulfilling demand industry requirements', and it is suggested to provide specific data support.
- The comparison between MOS and EC sensors on page 5 lacks quantitative data support. It is suggested to supplement specific indicators such as response time and sensitivity.
- The statement 'only a moderate group of authors have been dedicated to alleviate...' mentioned on page 9 is not coherent, and it is recommended to rewrite it.
- The conclusion section provides vague suggestions for future directions and suggests proposing more practical research paths.
- On page 6, there is a sudden transition from the sensor structure to the calibration process, which is somewhat abrupt. It is recommended to add a connecting sentence.
- Page 12 jumps from environmental factor research to early fire detection, lacking logical bridges between paragraphs.
Author Response
- There are some grammatical and expression issues that require further refinement.
(Response): We will check on the grammatical and expression issues but would appreciate if you can point out the issues in specific. Thanks for pointing out.
2. The conclusion of the abstract mentions 'well suited with', it is suggested to change it to 'well suited to'.
(Response): We will amend accordingly
3. The introduction mentions' falling short of fulfilling demand industry requirements', and it is suggested to provide specific data support.
(Response): Can you please explain what is "fulfilling demand industry requirements"?
4. The comparison between MOS and EC sensors on page 5 lacks quantitative data support. It is suggested to supplement specific indicators such as response time and sensitivity.
(Response): MOS and EC sensors are two different technologies. Page 5 serves as a brief introduction for these two common sensor technologies used in Semiconductor wafer fab. The cited statement "In comparison of the response time between the EC and MOS sensors, MOS sensors have a faster response time, but are prone to cross sensitivities of solvent vapors" are generalized characteristic of these sensor. The generalized characteristic of these sensors are cited from published papers or books.
5. The statement 'only a moderate group of authors have been dedicated to alleviate...' mentioned on page 9 is not coherent, and it is recommended to rewrite it.
(Response): We will look into this.
6. The conclusion section provides vague suggestions for future directions and suggests proposing more practical research paths.
(Response): We concluded the past experiments did not reflect the actual application of gas sensors in industrial setting. The data collected and model developed are not replicable in the real world especially in the context of Semiconductor wafer fab. That's why we have suggested "developing application-specific datasets that reflect actual environmental and operational
conditions". For example, further researches or experiments can be conducted to collect VOC gas or solvent gas data from the fume hood in the laboratories within Semiconductor wafer fab. Researcher can choose to install gas sensors at the fume hood and exhaust ducting (with measured DP, airflow, humidity, temperature for data collection) or the researcher can reconstruct the fume hood and ducting in any available space to mimic the condition in the laboratories. This is application specific in comparison with Wind Tunnel or any other experiments cited in the paper. It is practical for the researchers who have knowledges in industrial setting.
7. On page 6, there is a sudden transition from the sensor structure to the calibration process, which is somewhat abrupt. It is recommended to add a connecting sentence.
(Response): There are transition sentences to transition the "sensor structure" or rather "sensor technologies" => In
order for the gas detector to function according to the manufacturer’s design standard, periodic maintenance is required, such as sensor replacement and calibration. Routine calibration and bump test can check the gas sensor depletion rate which is subjective to the application and environment where the gas detectors are installed.
8. Page 12 jumps from environmental factor research to early fire detection, lacking logical bridges between paragraphs.
(Response): The main discussion on page 12, 4.3 Application-Specific Deployments and Environmental Variable Studies mainly discussing the effects of environmental factor (external interferences) on the gas sensor performance in differentiating the targeted gas from interferences. UAV's rotor speed induced draft (interference) on gas sensor measuring NO2 (target gas), Photoionization Diode in monitoring Benzene (target gas) vs Volatile Organic Compound (interference), MOX sensor on APR targeting gas leak (target gas) vs other vapors (interference).
The researches on early fire detection are relevant because the experiments conducted involved an array of sensors detecting smoldering fire that generated gases such as CO and VOCs before smoke. The development and application of algorithms to differentiate the gas sensing data, the target gases (CO & VOCs) vs interference gases.
Reviewer 2 Report
Comments and Suggestions for Authors
Overall, the review was collective and presented neatly. However, I suggest addressing appended comments to enhance the work.
1) "This systematic review recent research aimed at minimizing or eliminating the dependency on manual calibration." There are redundant words like "recent"
2) "These facilities are also highly process intensive and involve the use of specialty gases that are flammable, combustible, and hazardous to both human health and the environment" I suggest adding the flammable, combustible gases used in the fabrication of semiconductor wafer.
3) "However, these sensors can also respond to interference gases with similar chemical characteristics, which can trigger false alarms, cause unnecessary evacuations, and cause costly production shutdowns". I suggest citing and discussing reported works on the impact of interfering gases on sensing behavior.
4) Include a few reported works on e-nose on the detection of potential gases and its shortcomings apart from calibration transfer.
5) Fig.3: Format the table
6) L167: subscript
7) L169: subscript
8) L171: subscript
9) Place Fig. 7 below Fig 6
10) Improve the resolution of Fig.7
11) "Gas detectors from different manufacturers have different performance specification, maintenance procedure, and characteristic" The statement is incomplete.
12) Why was the literature search not conducted using Web of Science?
13) Increase the dimensions of fig.9 for better readability.
14) Format Fig.11
15) L437: subscript
16) I suggest including a table consisting of sensor materials, gas type, and its limit of detection.
17) Reference: I do not see reported works in 2025. It must be reworked. Around 4-5 references are from 2024.
18) Fix minor grammatical errors throughput the manuscript.
Author Response
1) "This systematic review recent research aimed at minimizing or eliminating the dependency on manual calibration." There are redundant words like "recent"
(Response) This systematic review recent research aimed at minimizing (updated)
2) "These facilities are also highly process intensive and involve the use of specialty gases that are flammable, combustible, and hazardous to both human health and the environment" I suggest adding the flammable, combustible gases used in the fabrication of semiconductor wafer.
(Response) I would appreciate if you can clarify further whether did you mean to list down the flammable, combustible gases in this paragraph? Or you have suggested to change the sentence? In a semiconductor wafer fab environment, the combustible or flammable gases are mostly hydrogen and some methane. Specialty gasses are listed in Figure 3 (page 3)
3. "However, these sensors can also respond to interference gases with similar chemical characteristics, which can trigger false alarms, cause unnecessary evacuations, and cause costly production shutdowns". I suggest citing and discussing reported works on the impact of interfering gases on sensing behavior.
(Response) Well noted. Will cite and discuss reported work in the revision.
4. Include a few reported works on e-nose on the detection of potential gases and its shortcomings apart from calibration transfer.
(Response) Well noted. We will include some reported works on e-nose.
5-10. Fig.3: Format the table to Improve the resolution of Fig.7
(Response) We will do our best to tidy up and improve the resolution of Figure 7.
11. "Gas detectors from different manufacturers have different performance specification, maintenance procedure, and characteristic" The statement is incomplete.
(Response) This is a complete statement. Can you please clarify which part of it is incomplete?
12) Why was the literature search not conducted using Web of Science?
(Response) We use the platforms that are accessible through Multimedia University, Cyberjaya.
13) Increase the dimensions of fig.9 for better readability.
(Response) Well noted. Will amend it accordingly.
14) Format Fig.11
(Response) Well noted.
15) L437: subscript
(Response) Well noted
16) I suggest including a table consisting of sensor materials, gas type, and its limit of detection.
(Response) This could be challenging. Gas sensors that are widely used in Semiconductor wafer fab are Electrochemical (EC) and Semiconductor sensor (SG). These are sensor type or sensor technologies. Different manufacturers will have their variance or model using commonly known material doped or mixed with proprietary or undisclosed material. These sensors are designed and made for specific gases, limit of detection or detection range.
17) Reference: I do not see reported works in 2025. It must be reworked. Around 4-5 references are from 2024.
(Response) We conducted the literature search and review early in 2025. This is the reason why no reported works in 2025.
18) Fix minor grammatical errors throughput the manuscript.
(Response) Well noted and will double check and get the errors fixed. We were surprise Overleaf LaTex didn't pick up errors. We subscribed to their auto correction and proof reading features.
Thanks for reviewing the manuscripts.
Reviewer 3 Report
Comments and Suggestions for Authors
By starting and strongly linking with the wafer fabrication industry, the manuscript “Autonomous Hazardous Gas Detection Systems: A Systematic Review” contributes to unique technical knowledge, unlike many review articles on gas sensors. It is commendable that the authors provided content outside academic sources and then performed a systematic review highlighting relevant development. Only minor revision is required before this manuscript can be accepted for publication.
1 First of all, the title is rather generic and does not reflect the uniqueness of the review content. The authors should consider adding the term “industry” or others.
2 Figures are interesting but some of them are not cited in the text. Captions should be improved. Figures 10 and 11 must be presented as Tables. Please correct these issues.
3 The outline of “Section 5 The Research Gap and Future Research Direction” is not quite right. It should be restructured 5.1, 5.2, 5.3 as 5.2.1, 5.2.2, 5.2.3, while keeping Subsection 5.1 for describing Research Gap.
4 The conclusion section is currently bloated and needs to be rewritten for clarity and conciseness. For a review article with research outlook section, the issues and implications can be highlighted in a more succinct manner.
5 While the article is generally well written, some sentences need revision, e.g., A primary issue lies in the very foundation of algorithmic development: datasets.
6 The authors should keep consistency throughout the manuscript, i.e., the use of capital letters in the reference list and the punctuation with spacebar must not be random.
Author Response
1 First of all, the title is rather generic and does not reflect the uniqueness of the review content. The authors should consider adding the term “industry” or others.
(Response) We will look into this. Thanks for the comment.
2 Figures are interesting but some of them are not cited in the text. Captions should be improved. Figures 10 and 11 must be presented as Tables. Please correct these issues.
(Response) We will amend accordingly.
3. The outline of “Section 5 The Research Gap and Future Research Direction” is not quite right. It should be restructured 5.1, 5.2, 5.3 as 5.2.1, 5.2.2, 5.2.3, while keeping Subsection 5.1 for describing Research Gap.
(Response) We will look through the section and subsection and amend accordingly.
4 The conclusion section is currently bloated and needs to be rewritten for clarity and conciseness. For a review article with research outlook section, the issues and implications can be highlighted in a more succinct manner.
(Response) Your comments are well noted. We will review it and elaborate with improved clarity.
5 While the article is generally well written, some sentences need revision, e.g., A primary issue lies in the very foundation of algorithmic development: datasets
(Response) Revision "A primary issue lies in the generation of datasets which is the very foundation of algorithmic development."
6 The authors should keep consistency throughout the manuscript, i.e., the use of capital letters in the reference list and the punctuation with spacebar must not be random.
(Response) The wordings in the reference list are directly copied from the citation provided for BibTex LaTex for example:
@inproceedings{chambers2016managing,
title={Managing hazardous process exhausts in high volume manufacturing (HVM) of advanced devices},
author={Chambers, Andrew},
booktitle={2016 China Semiconductor Technology International Conference (CSTIC)},
pages={1--3},
year={2016},
organization={IEEE}
}
(Response) We do appreciate your feedbacks and comments. We will do our best for the revision.
Reviewer 4 Report
Comments and Suggestions for Authors
COMMENTS TO AUTHORS:
The manuscript reviewed the recent Gas Detection Systems (GDS) with minimum or eliminated dependency on manual calibration. The whole paper made a comprehensive discussion on the hazardous gases, Calibration of GDS, etc. and provided the discussion on advanced machine learning based Calibration Methodology. This paper is interesting and carries substantial importance in the gas-related sensing systems. I recommend the acceptance of this manuscript after with some revisions. The detailed suggestions can be checked as follows:
- The author mentioned in Page 2, Line 51-57: “In a typical wafer fab facility, there may be hundreds or thousands of gas detectors, making the calibration process highly resource intensive. Reducing the number of installed detectors is generally not viable because of the complexity and diversity of the specialty gases used.” What are the commonly used strategies for the calibration of those hundreds or thousands of gas detectors and how complex are those traditional calibration processes?
- On page 8 for data-based calibration, in the study selection and data extraction part, if the size of database used in this step affect the final results? How do they control this reason caused difference?
- The minimum detection limit is an important factor for evaluating gas sensors, it might be better if the author make a further discussion on this point.
Author Response
- The author mentioned in Page 2, Line 51-57: “In a typical wafer fab facility, there may be hundreds or thousands of gas detectors, making the calibration process highly resource intensive. Reducing the number of installed detectors is generally not viable because of the complexity and diversity of the specialty gases used.” What are the commonly used strategies for the calibration of those hundreds or thousands of gas detectors and how complex are those traditional calibration processes?
(Response) Thanks for pointing out. In the initial manuscripts we did discuss the process of calibration but eventually decided to remove it due to the length of the manuscripts. We have included a brief overview of calibration process that has been a common practice in the industrial setting. The standard manual gas detector calibration process entails a trained professional utilizing a portable calibration gas cylinder, injecting the gas into the detector, and subsequently adjusting the detector’s reading to align with the known concentration of the calibration gas.
2. On page 8 for data-based calibration, in the study selection and data extraction part, if the size of database used in this step affect the final results? How do they control this reason caused difference?
(Response) We have tried numerous way to search the database using individual key words such as "gas sensor", "calibration" "accuracy drift" and etc. at the initial stage. The outcome of the search was overwhelming and out of context. We eventually refined our search concepts to include "gas sensor", "chemical sensor,"
"e-nose," "calibration", "drift compensation", "self-calibration," and "calibration transfer"
and "AI" and "machine learning". These terms were combined using Boolean operators
("AND," "OR") to achieve high sensitivity.
3. The minimum detection limit is an important factor for evaluating gas sensors, it might be better if the author make a further discussion on this point.
(Response) The minimum detection limit is indeed important factor for evaluating gas sensor. However, the objective of our research is not to evaluate gas sensor but to study the long term performance of the gas sensors based on the specific industrial application. Once we have collected sufficient data from the gas sensors in the specific conditions of the industrial application, we can develop algorithm(s) to predict the accuracy drift. Once the algorithm has been developed, the manual calibration will not be necessary in the future. Cross sensitivity or interference gases that triggered false alarm is another major topic in the industrial world. We have reviewed some of the past researches on E nose and smoldering fire detection to address this issue. We will similarly address this subject in an industrial setting experiment.
(Response) We have well noted on your feedbacks and comments. We will do our best in the revision. Thank You.
Round 2
Reviewer 1 Report
Comments and Suggestions for Authors
The author has carefully revised the paper and provided detailed responses to the reviewers' comments. The quality of the paper has significantly improved, and I believe it can be accepted.